# Factors associated with comprehensive knowledge of antenatal care and attitude towards its uptake among women delivered at home in rural Sehala Seyemit district, northern Ethiopia: A community-based cross-sectional study

**Azmeraw Ambachew Kebede**[1]*, **Birhan Tsegaw Taye**[2], **Kindu Yinges Wondie**[1]

1 Department of Clinical Midwifery, School of Midwifery, College of Medicine and Health Sciences, University of Gondar, Gondar, Ethiopia, 2 School of Nursing and Midwifery, Asrat Woldeyes Health Science Campus, Debre Berhan University, Debre Berhan, Ethiopia

* azmuzwagholic@gmail.com

## Abstract

### Background

Despite the current sustainable development goal period (i.e. SDG 3), the prevention of maternal and neonatal mortality is not satisfactory in Ethiopia. Keeping women knowledge-able about antenatal care and maintaining a positive attitude towards its uptake, particularly in the marginalized rural community is crucial. However, evidence regarding the knowledge and attitude of women towards antenatal care uptake is scarce. Therefore, this study aimed to assess factors affecting comprehensive knowledge of antenatal care and attitude towards its uptake among women delivered at home in rural Sehala Seyemit district, northern Ethiopia.

### Methods

A community-based cross-sectional study was conducted from September 1st to October 15th, 2020. A multi-stage sampling technique was used to select 653 women. The data were collected using a semi-structured interview guide. Data were entered into EPI INFO 7.1.2 and analyzed by SPSS version 25. Both bivariable and multivariable logistic regression analyses were undertaken to identify factors associated with women's knowledge of antenatal care and attitude towards its uptake. The level of significant association in the multivariable analysis was determined based on a p-value of < 0.05.

### Results

Women's knowledge of antenatal care and positive attitude towards its uptake was 56.5% and 75.2%, respectively. Older age (AOR = 7.2; 95% CI: 3.43, 15.1), media exposure (AOR = 3.69; 95% CI: 2.41, 5.65), history of abortion (AOR = 11.6; 95% CI: 3.3, 14.6), time to

**Data Availability Statement:** All relevant data are within the manuscript and it's supporting information files.

**Funding:** The authors received no specific funding for this work.

**Competing interests:** The authors have declared that no competing interests exist.

**Abbreviations:** AOR, adjusted odds ratio; ANC, antenatal care; CI, confidence interval; COR, crude odd ratio; HEWs, health extension workers; MMR, maternal mortality ratio; SDG, sustainable development goal; SPSS, statistical package for social science; SSA, Sub-Saharan Africa; TBA, traditional birth attendants; VIF, variance inflation factors; WHO, world health organization.

reach health facility (AOR = 4.58; 95% CI: 3.05, 6.88), and history of obstetric danger signs (AOR = 7.3; 95% CI: 3.92, 13.64) were factors significantly associated with knowledge of antenatal care. Furthermore, higher decision-making power (AOR = 8.3; 95% CI: 4.8, 13.83), adequate knowledge of antenatal care (AOR = 2.2; 95% CI: 1.26, 3.71), delivery attended by health extension workers (AOR = 2.3; 95% CI: 1.1, 5.1), and media exposure (AOR = 2.27; 95% CI: 1.30, 3.97) were predictors of a favorable attitude towards antenatal care utilization.

## Conclusion

Although the majority of women in the present study had a favorable attitude towards antenatal care uptake, their knowledge level was inadequate. Strengthening access to transportation, mass media, involvement in household decision-making, and encouraging women to deliver at a health facility by a skilled provider may increase women's knowledge and attitude towards antenatal care uptake, thereby improving maternal healthcare service uptake.

## Introduction

Skilled care during pregnancy, delivery and the postpartum period are essential interventions that ensure favorable maternal and neonatal outcomes [1]. Of these, antenatal care (ANC) is the care provided to all pregnant women by qualified health professionals during pregnancy [2]. It is given to all pregnant women regardless of prerequisites. However, the number of ANC contacts might increase in women with medical problems. Antenatal care aids pregnant women and their families to become arranged and prepared for the consequences related to pregnancy and childbirth [3]. Accordingly, the primary purpose of ANC is to ascertain pregnant women who are at risk, support them to be healthy, prevent disease, and improve healthcare access throughout the continuum of maternal care [4].

Antenatal care gives great opportunities to keep pregnant women healthy and decreases the hazards of pregnancy adverse events and neonatal mortality, and morbidity [5]. Besides, evidence supports that ANC enhances women's knowledge of maternal and neonatal danger signs [6,7], a roadmap for health facility delivery [8], and increases postnatal care utilization [9]. Despite these advantages, many women in rural areas are still not able to use maternal health services and prefer giving childbirth at their home [10,11]. Inadequate knowledge of women about maternal health service utilization could be listed as a prior reason for the low utilization of health services. As a result, building up women's knowledge of ANC and attitudes towards its uptake can drop the tragic maternal mortality and morbidity. Because sufficient knowledge of ANC can increase the uptake of maternal health services (i.e. optimum ANC, health facility delivery, and postnatal care utilization) [12]. Thus, the care provided during pregnancy is a precondition for the eventual reduction of maternal, neonatal, and perinatal mortalities and morbidities associated with pregnancy and childbirth [13].

In 2016, the World Health Organization (WHO) recommends that all mothers should have at least eight ANC contacts to ensure better maternal and neonatal health [13]. However, the 4 ANC is still implemented in Ethiopia. In this context, about 295,000 women died from pregnancy and childbirth-related complications in 2017 Globally [14]. Despite the current sustainable development goal (i.e. SDG 3) period where great emphasis has been given to maternal and neonatal health, ensuring the optimal health of both the mother and newborns is the

foremost challenge in Sub-Saharan Africa (SSA) [15,16]. Seven out of ten women died from SSA [14]. In addition, Ethiopian institutional birth is one of the lowest in the world. Fifty-two percent of women gave birth at home, indicating a significant number of deliveries continue to take place at home. Moreover, there is a disparity in service utilization between urban (72%) and rural (43%) [1]. Since no healthcare system allows professionals to attend labor at home, homebirth in Ethiopia is unattended, unless by Traditional Birth Attendants (TBA) [17]. According to the Ethiopian Health Sector Transformation Plan II report, the maternal mortality ratio (MMR) was 401/100,000 live births in 2017. In addition, infant mortality per 1000 live births reduced from 77 in 2005 to 47 in 2019 [18]. However, over the years, there have been no significant reductions in neonatal mortality [1]. This high rate of mortality is due to poor use of health services.

The proportion of women in Ethiopia who received ANC from a skilled provider has increased from 27% in 2000 to 62% in 2016, and 32% of women had at least four ANC visits during their last pregnancy in 2016 [19]. Similarly, the 2019 Ethiopian mini demographic health survey concluded that 57% of women did not receive optimum ANC in their recent pregnancy [1]. This implies that there is a need to do more to increase ANC utilization.

Various studies have assessed the utilization and factors affecting ANC in Ethiopia [3,20–27]. However, a few studies were conducted on women's knowledge of ANC and attitudes towards its uptake [4,28]. It is evidenced that adequate knowledge of ANC increases pregnant women's adherence to supplemental drugs (for example, folic acid, vitamins, and de-worming) and ability to identify danger signs of pregnancy, and take an action promptly [29]. Besides, the provision of quality maternal healthcare services helps to prevent and manage pregnancy-related maternal and neonatal complications and deaths [30–32].

Studies described that women's knowledge of antenatal care was 79.2% in Ghana [5], 92.3% in Saudi Arabia [29], and 85.3% in Libya [33]. Evidence in Ethiopia also revealed that knowledge of ANC was 88.2% [4] and 90.7% [28]. Besides, findings showed that the attitude of women towards antenatal care uptake was 90.3% in Saudi Arabia [29], 96.0% in Libya [33], and 69.6% in Mizan, Ethiopia [4]. However, most of these studies addressed women who have already engaged in maternity health services, disregarding the rural community, particularly women who gave childbirth at home. In connection to this, there is a need to assess the knowledge and attitude of the conservative rural community towards ANC utilization to be in line with the WHO target, which says 90% of pregnant women should have at least four and above ANC contacts by 2025 towards increasing to 8 contacts by 2030 [34]. It is well known that the practice of institutional delivery is lower in the rural community, and women's awareness of maternal health services is expected to be limited. Moreover, previous studies didn't include important variables that would affect women's knowledge and attitude towards ANC, such as household decision-making power and husband involvement.

Increasing women's knowledge of ANC and attitude towards its uptake is the pillar to expanding maternity and child healthcare utilization throughout the maternal continuum of care. Over the last decade, knowledge and attitude towards ANC have changed around the world and the concept of fit pregnancy steadily gained popularity based on its positive outcomes [35–38]. To improve women's knowledge of ANC and attitude towards its utilization, the target group should be those women who gave birth at their home, which will be positively associated with a facility birth utilization. However, evidence in this regard is scarce in Ethiopia, particularly in the study area. Therefore, this study will fill the gaps and assess the factors affecting comprehensive knowledge of ANC and attitude towards its uptake among women who gave birth at home in rural Sehala Seyemit district, northern Ethiopia. Besides, doing the research may offer basic evidence for any interventions aimed at improving maternal health and preventing perinatal mortality and morbidity.

## Methods and materials

### Study design, setting, and period

A community-based cross-sectional study was conducted from September 1st to October 15th, 2020. This study was conducted in rural Sehala Seyemit district, Waghimra zone, Amhara regional state, Northern Ethiopia. Sehala Seyemit district is located 285 km northeast of Bahir Dar (the capital city of Amhara regional state) and about 799 km north of Addis Ababa (the capital city of Ethiopia). Accessing health services in the district is difficult because of the lack of transportation to each *"kebeles"* (which is the smallest administrative unit in Ethiopia). Maternal health services such as ANC, childbirth, and postnatal care are given for free in Ethiopia, including the study area. The district has 13 *"kebeles"*; 12 rural and 1 urban "kebeles". Currently, the district has a population of 39,435. Over 90% of the population are farmers. Moreover, there are 3 health centers and 13 health posts serving the community. Furthermore, there were a total of 923 women who gave birth at home in the last 2 years (Sehala Seyemit Woreda report, unpublished data).

### Study population

All women who gave birth in the last two years in the selected *"kebeles"* during the data collection period were selected to be part of the study. All critically ill women throughout the data collection period were excluded.

### Sample size determination and sampling procedure

The sample size for this study was determined by using a single population proportion formula by considering the following assumptions: women's attitude towards ANC in Mizan, Ethiopia-70.6% [4], 95% level of confidence, and 5% margin of error. Thus, $n = \frac{(Z\alpha/2)^2 p(1-p)}{d^2} = n = \frac{(1.96)^2 * 0.706(1-0.706)}{(0.05)2} = 319$. Where, n = required sample size, $\alpha$ = level of significant, z = standard normal distribution curve value for 95% confidence level = 1.96, p = women's attitude towards ANC, and d = margin of error. By considering a design effect of 2 (since multistage sampling) and a 5% non-response rate, the minimum adequate sample size was 670. A multistage sampling technique was employed to select the study participants. In the first stage, eight kebeles were selected randomly among the 12 rural *"kebeles"*. In addition, the lists of home-delivered women from the selected kebeles were obtained from health extension workers (HEWs) and local administrators. Thereafter, the sampling frame was designed by numbering the list of women. Then, the total sample size was distributed to each selected *"kebeles"* proportionally. In the second stage, the women were selected by a simple random sampling technique using a table of random generation.

### Variables of the study

Women's knowledge of ANC (adequate/inadequate) and women's attitude towards ANC uptake (favorable/unfavorable) were the outcome variables. Whereas age of the women, marital status, women's educational status, women's occupation, husband educational status, husband occupation, family size, exposure to mass media, time to reach the nearby health facility, parity, history of ANC, number of ANC, birth assistant, husband involvement in maternal and children's health, household decision-making power, history of abortion, history of neonatal death, history of obstetric danger signs during pregnancy, and status of the pregnancy were the explanatory variables.

## Measurements

**Home delivery:** Is defined as a birth that has taken place at the laboring woman's own home, or her relative, or her neighbor rather than a birthing center without a skilled birth attendant [39].

**Traditional birth attendant:** Is a person who is traditionally experienced in attending labor to assist women in childbirth and give care during pregnancy and childbirth [38].

**Comprehensive knowledge of ANC:** Includes knowledge of ANC and pregnancy, knowledge of obstetric danger signs, knowledge of birth preparedness and complication readiness, knowledge of malaria prevention, knowledge of anemia prevention, knowledge of helminthic infection prevention, and knowledge of tetanus prevention during pregnancy. A total of 20 open and close-ended questions were designed to assess the comprehensive knowledge of ANC. Correct and/or "Yes" answers were coded as 1, whereas incorrect and/or "No"/don't know answers were coded as 0. The minimum and maximum scores were 0 and 20, respectively. Thus, based on the summative score of variables designed to assess knowledge, a score above the mean was considered knowledgeable [4,5,40] **(S1 File).**

**Women's attitude:** Women's attitude towards ANC was measured using 9 questions: 1) Want to have ANC follow up for next time 2) Intention to deliver in a health facility for the next pregnancy 3) Health care professionals providing antenatal care is good 4) All pregnant women should have ANC follow up 5) Timely ANC follow up will be safer for both mother and baby 6) Want to pay for ANC if it is with fee 7) Husbands should be present during ANC follow-up 8) Advice regarding proper health during pregnancy can be gotten outside the hospital 9) Follow up during pregnancy may decrease antenatal and postnatal complications. Each question has a five-point Likert scale (1 = strongly disagree, 2 = disagree, 3 = neutral, 4 = agree, 5 = strongly agree). The total score was 9–45 and women who scored above the mean value were considered as having a favorable attitude [33,41,42].

**Husband involvement:** Husband involvement in maternal and child health-related activities was measured using 9 questions: 1) did your husband go with you for ANC follow-up at least once in your most recent pregnancy? 2) Did your husband provide transport/gave money for transport during your recent pregnancy or delivery? 3) Did your husband accompany you to the hospital during labor for your recent delivery? 4) Did your husband discuss with health care providers during your recent pregnancy or delivery? 5) Did your husband look after the child at home/stay with the babies while you are outside the home? 6) Did your husband bathe a newborn/infant while you are busy? 7) Did your husband buy clothes/other things for infants/neonates? 8) Did your husband go with you for immunization services? 9) Did your husband assist you while you breastfeed the newborn/infant? Each question was coded as 0 for "no" and 1 for "yes". The total score ranged from 0–9 and a score of above the mean was considered as husband involved Based on the summative score of variables designed to assess husband involvement a score above the mean was considered as involved [43,44].

**Household decision-making power:** Women's decision-making power was assessed using 9 questions: 1) who decides about health care for you? 2) Who decides on the large household purchase or sell? 3) Who decides on intra-household resource allocation/ daily household purchases? 4) Who decides on where and when to seek medical care for sick newborns/children? 5) Who decides on visits of family, friends, or relatives? 6) Who decides when to have an additional child? 7) Who usually decides how your partner's/husband earnings will be used? 8) Who decides to go for an ANC visit, postnatal (PNC) visit, where to deliver, and infant immunization? 9) Who usually decides what foods to be cooked each day? The possible answers were me alone which was coded as 2, both of us which was coded as 1, and the husband alone

or others which was coded as 0. The score ranged from 0 to 18 and a woman who scored above the mean was considered as having higher household decision-making power [45].

**Media exposure:** Those women who responded at least once a week to one of the media are considered to be regularly exposed to that form of media (i.e. TV, radio, or magazine) [19].

**Experienced danger signs:** Women who have experienced one or more of the danger signs during their last pregnancy or childbirth were considered as experienced danger signs.

**History of neonatal death**: Women who have experienced the death of a neonate within the first 28 completed days of life [46].

**History of abortion**: Women who have experienced termination of pregnancy before 28 weeks gestation in the Ethiopian context [47].

## Data collection instruments, procedures, and quality control

The data collection tool was developed by reviewing the literature [4,5,28,33,41,42]. The data were collected using a semi-structured interviewer-administered questionnaire through face-to-face interviews. Initially, the questionnaire was prepared in English and translated to the Amharic language, and back to English to ensure consistency. The questionnaire contains socio-demographic characteristics, reproductive and maternity healthcare characteristics, husband involvement in maternal and child health-related activities, household decision-making power, and questions assessing women's comprehensive knowledge of ANC, and attitude towards its uptake. The questionnaire was assessed by a group of researchers (three in the field of maternal and child health, one in the field of public health, and two midwives in the hospital) to evaluate and enhance the items in the question. Before the actual data collection, we did a pretest on 34 women at Ziquala Woreda which has similar socio-cultural and living standards as the study population. Eight female HEWs and four male Diploma in midwifery holders were recruited for data collection and supervision, respectively. Two days of training were given regarding the overall data collection process. During the data collection, the questionnaire was checked for completeness daily by the supervisors.

## Data processing and analysis

Data were checked, coded, and entered into EPI INFO version 7.1.2, and were exported to SPSS version 25 for further cleaning and analysis. Before analysis, re-coding, transforming, computing, and categorizing of variables were done. Descriptive statistics were used to show participants' characteristics, comprehensive knowledge of ANC, and attitude towards its uptake. Binary logistic regression analysis was fitted to identify statistically significant independent variables, and variables having a p-value of < 0.25 were included in the multivariable logistic regression for controlling confounders. The multicollinearity assumption was checked using the variance inflation factor (VIF), where VIF <10 was acceptable. In the multivariable logistic regression (Backward Likelihood Ratio approach), a p-value of < 0.05 with a 95% CI for the adjusted odds ratio was employed to ascertain the significant association.

## Ethical considerations

We conducted the study under the declaration of Helsinki. Ethical approval was obtained from the Institutional Ethical Review Board of Debre Berhan University (**protocol number; P005/20**). A formal letter of administrative support was gained from the Sehala Seyemit Woreda health office. Both oral and written informed consent was collected from each of the study members after a clear explanation of the aim of the study and their right to withdraw from the study at any time. For those women who cannot read and write, a thumbprint was taken (i.e. it is a common practice in Ethiopia including banking services and other large issues that need a

signature). The study participants were assured that the collected information is anonymous and kept confidential for the study purpose only.

## Result

### Socio-demographic characteristics of study participant

In this study, a total of 670 respondents were interviewed. Seventeen respondents were excluded for their incomplete data, giving a 97.5% response rate. The mean age of the respondents was 26.4 years (SD ±4.93) and more than two-thirds of the respondents were between the age group of 21–30 years. Most (97.2%) of the study participants were currently married and 384 (58.8%) of the study participants were unable to read and write. Over four-fifths of the women were farmers by occupation. Concerning the occupation of the husband, 85% were farmers and slightly more than half (51%) of them were unable to read and write (**Table 1**).

### Reproductive history and maternity healthcare service-related characteristics

Of the total study participants, only 198 (30.3%) women had at least one ANC visit in their recent pregnancy. About 58% of the participants were assisted by HEWs for their recent

**Table 1. Socio-demographic characteristics of study participants in rural Sehala Seyemit district, northern Ethiopia, 2020 (n = 653).**

| Characteristics | Category | Frequency | Percentage (%) |
|---|---|---|---|
| Age of women in year | ≤ 20 | 70 | 10.7 |
| | 21–30 | 440 | 67.4 |
| | ≥ 31 | 143 | 21.9 |
| Current marital status | Married | 635 | 97.2 |
| | Unmarried | 18 | 2.8 |
| Family size | 2–3 | 84 | 12.9 |
| | 4–6 | 433 | 66.3 |
| | ≥ 7 | 136 | 20.8 |
| Women's literacy | Unable to read and write | 384 | 58.8 |
| | Able to read and write | 254 | 38.9 |
| | Primary education | 15 | 2.3 |
| Women's occupation | Famer | 533 | 81.6 |
| | Merchant | 89 | 13.6 |
| | Others [a] | 31 | 4.8 |
| Husband literacy (n = 635) | Unable to read and write | 235 | 37 |
| | Able to read and write | 324 | 51 |
| | Primary education and above | 76 | 12 |
| Husband occupation (n = 635) | Famer | 540 | 85 |
| | Merchant | 92 | 14.5 |
| | Others [a] | 3 | 0.5 |
| Exposure to mass media | Yes | 433 | 66.3 |
| | No | 220 | 33.7 |
| Type of transport at the time of emergency | On foot/ traditional ambulance | 426 | 65.3 |
| | Ambulance | 142 | 21.7 |
| | Public transport | 85 | 13 |

Note

[a] student and daily labor, traditional ambulance- it is made of wood and is used to transport mothers to health facilities in areas where there is no car.

**Table 2. Reproductive and maternity healthcare service characteristics of the study participants in rural Sehala Seyemit district, northern Ethiopia, 2020 (n = 653).**

| Characteristics | Category | Frequency | Percentage (%) |
|---|---|---|---|
| **Parity** | 1 | 74 | 11.3 |
| | 2–4 | 436 | 66.8 |
| | ≥5 | 143 | 21.9 |
| **Had ANC** | Yes | 198 | 30.2 |
| | No | 455 | 69.8 |
| **Number of ANC follow-up (n = 198)** | <4 | 191 | 96.5 |
| | ≥4 | 7 | 3.5 |
| **Birth assistant** | HEWs | 379 | 58 |
| | TBA | 204 | 31.3 |
| | Family members | 70 | 10.7 |
| **Had postnatal care** | Yes | 82 | 12.6 |
| | No | 571 | 87.4 |
| **Husband involvement** | Involved | 363 | 56.6 |
| | Not involved | 290 | 44.4 |
| **History of abortion** | Yes | 63 | 9.6 |
| | No | 590 | 90.4 |
| **History of neonatal death** | Yes | 19 | 2.9 |
| | No | 634 | 97.1 |
| **Experienced obstetric danger signs** | Yes | 119 | 18.2 |
| | No | 534 | 81.8 |
| **Time to the health facility** | < 1 hour | 326 | 49.9 |
| | ≥ 1 hour | 327 | 50.1 |
| **Household decision-making power** | Higher | 495 | 75.8 |
| | Lower | 158 | 24.2 |
| **Pregnancy status** | Planned | 573 | 87.7 |
| | Unplanned | 80 | 12.3 |

delivery. More than half (56.6%) of women got their husband's involvement in maternity and childcare-related activities (**Table 2**).

## Women's comprehensive knowledge of ANC

Overall, 56.5% of women had adequate knowledge of antenatal care. The majority (85.1%) of respondents ever heard about ANC. More than half (56.2%) of women knew the importance of early ANC visits. Only 17.9% of women had known the recommended number of ANC (**Table 3**).

## Participant's attitude toward ANC uptake

Three-fourths (75.2%) of women had a favorable attitude towards ANC uptake. About 148 (22.7%) women strongly agreed that husbands should present during ANC visits. Only11.6% of women had a strong desire to pay for ANC if it is with a cost (**Table 4**).

## Barriers towards using ANC and institutional delivery

All women who were part of this study gave their recent birth at home. Besides, the majority of women didn't have ANC visits for their recent pregnancies. The main reasons mentioned by the study participants for not having ANC and institutional delivery were long travel time and/or lack of transportation, considering ANC may not be necessary, and having a workload (**Fig 1**).

**Table 3. Comprehensive knowledge of antenatal care among women delivered at home in rural Sehala Seyemit district, northern Ethiopia, 2020 (n = 653).**

| Variables | Frequency | Percentage (%) |
|---|---|---|
| Ever heard about ANC | Yes (556) | 85.1 |
| | No (97) | 15.9 |
| Starting early ANC is important | Yes (367) | 56.2 |
| | No (286) | 43.8 |
| Pregnant women may have problems without ANC | Yes (402) | 61.6 |
| | No (251) | 38.4 |
| ANC has to be recommended regardless of complications | Yes (382) | 58.5 |
| | No (271) | 41.5 |
| Maternal waiting homes are important in area's far from a health facility | Yes (357) | 54.7 |
| | No (296) | 45.3 |
| Health facility delivery is safer and better than home delivery | Yes (534) | 81.8 |
| | No (119) | 18.2 |
| Regular ANC medications can promote optimal growth of the unborn fetus | Yes (426) | 65.2 |
| | No (227) | 34.8 |
| ANC can prevent complications | Yes (452) | 69.2 |
| | No (201) | 30.8 |
| Alcohol drinking during pregnancy is bad for the fetus | Yes (329) | 50.4 |
| | No (324) | 49.6 |
| Smoking during pregnancy is unsafe for the fetus | Yes (585) | 96.9 |
| | No (68) | 3.1 |
| Do you know when to start ANC | Correct answer (362) | 55.4 |
| | Incorrect answer (291) | 44.6 |
| Perception of first fetal movement | Correct answer (271) | 41.5 |
| | Incorrect answer (382) | 58.5 |
| At which stage of pregnancy fetal deformities most likely occur? | Correct answer (124) | 19 |
| | Incorrect answer (529) | 81 |
| The recommended number of ANC | Correct answer (117) | 17.9 |
| | Incorrect answer (536) | 82.1 |
| Prevention of malaria during pregnancy | Correct answer (267) | 40.9 |
| | Incorrect answer (386) | 59.1 |
| Prevention of anemia during pregnancy | Correct answer (235) | 36 |
| | Incorrect answer (418) | 64 |
| Prevention of tetanus during pregnancy | Correct answer (194) | 29.7 |
| | Incorrect answer (459) | 70.3 |
| Prevention of intestinal parasitic infection during pregnancy | Correct answer (256) | 39.2 |
| | Incorrect answer (397) | 60.8 |
| Obstetric danger signs during pregnancy | Correct answer (239) | 36.6 |
| | Incorrect answer (414) | 63.4 |
| Components of birth preparedness and complication readiness plan | Correct answer (209) | 32 |
| | Incorrect answer (444) | 68 |
| What complications a woman will face without ANC? | Correct answer (376) | 57.6 |
| | Incorrect answer (277) | 42.4 |
| Overall comprehensive knowledge of ANC | Adequate knowledge (369) | 56.5 |
| | Inadequate knowledge (284) | 43.5 |

**Table 4. Attitude towards antenatal care uptake by component among women delivered at home in rural Sehala Seyemit district, northern Ethiopia, 2020 (n = 653).**

| Variables | Strongly agree (%) | Agree (%) | Neutral (%) | Disagree (%) | Strongly disagree (%) |
|---|---|---|---|---|---|
| Want to have ANC follow up for next time | 158 (24.2) | 295 (45.1) | 121 (18.5) | 48 (7.4) | 31(4.8) |
| Intention to deliver in a health facility for the next pregnancy | 85 (13) | 280 (42.9) | 108 (16.5) | 105 (16.1) | 75 (11.5) |
| Health care professionals providing antenatal care is good | 32 (4.9) | 382 (58.6) | 118 (18) | 84 (12.8) | 37 (5.7) |
| All pregnant women should have ANC follow up | 221 (33.8) | 260 (39.8) | 88 (13.5) | 53 (8.1) | 35 (5.4) |
| Timely ANC follow up will be safer for both mother and baby | 145 (22.2) | 305 (46.7) | 82 (12.6) | 95 (14.5) | 26 (4) |
| Want to pay for ANC if it is with fee | 76 (11.6) | 186 (28.5) | 91 (13.9) | 180 (27.6) | 120 (18.4) |
| Husbands should be present during ANC follow-up | 148 (22.7) | 370 (56.6) | 64 (9.8) | 41 (6.3) | 30 (4.6) |
| Advice regarding proper health during pregnancy can be gotten outside the hospital | 34 (5.2) | 154 (23.6) | 290 (44.4) | 105 (16.1) | 70 (10.7) |
| Follow up during pregnancy may decrease antenatal and postnatal complications | 135 (20.7) | 333 (51) | 85 (13) | 62 (9.5) | 38 (5.8) |
| Overall attitude | Favorable (491) | | | 75.2% | |
| | Unfavorable (162) | | | 24.8 | |

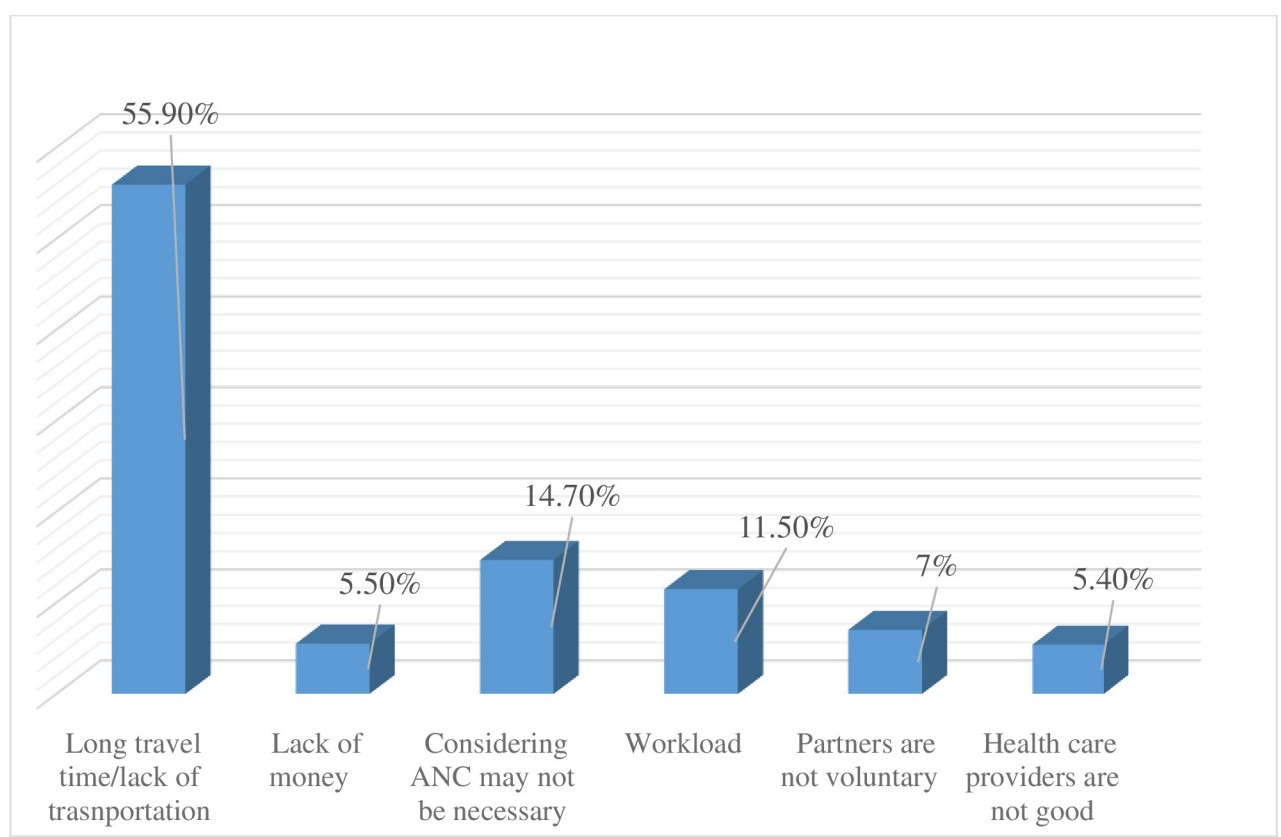

**Fig 1. Barriers to use antenatal care and institutional delivery among home-delivered women in rural Sehala Seyemit district, northern Ethiopia, 2020.**

## Factors associated with women's knowledge of antenatal care

Multivariable logistic regression analysis revealed that women's age, media exposure, history of abortion, had experienced obstetric danger signs in the recent pregnancy, and time to reach the health facility were factors significantly associated with women's comprehensive knowledge of ANC.

Women who had mass media exposure were 3.69 times more likely to have had adequate knowledge of ANC as compared to women who didn't have media exposure (AOR = 3.69; 95% CI: 2.41, 5.65). Women who have had a history of abortion were 11.6 times more likely to have had adequate knowledge of ANC as compared to their counterparts (AOR = 11.6; 95% CI: 3.3, 14.6). Similarly, the odds of having adequate knowledge of ANC among women who travel less than one hour to reach the health facility were 4.58 (AOR = 4.58; 95% CI: 3.05, 6.88) times higher as compared to women who travel for more than one hour. This study also found that women greater than 30 and 20–30 years old were seven (AOR = 7.2; 95% CI: 3.43, 15.1) and three (AOR = 3.26; 95% CI: 1.73, 6.15) times more likely to have had adequate knowledge of ANC when compared with women less than or equal to 20 years old respectively. Moreover, those women who have had a history of obstetric danger signs during their recent pregnancy were seven times (AOR = 7.3; 95% CI: 3.92, 13.64) more likely to have had adequate knowledge of ANC compared with those women who haven't had danger signs (**Table 5**).

**Table 5. Bi-variable and multivariable logistic regression analysis of factors associated with knowledge of antenatal care among women delivered at home in rural Sehala Seyemit district, northern Ethiopia, 2020 (n = 653).**

| Variables | Category | Knowledge of women | | COR (95% CI) | AOR (95% CI) | P-value |
|---|---|---|---|---|---|---|
| | | Adequate | Inadequate | | | |
| **Age in year** | $\leq 20$ | 31 | 39 | 1 | 1 | |
| | 21–30 | 219 | 221 | 1.25 (0.75, 2.07) | 2.26 (1.73, 6.15) | 0.032 |
| | >31 | 119 | 24 | 6.24 (3.3, 11.9) | 7.2 (3.43,15.1) | 0.040 |
| **Women's occupation** | Farmer | 286 | 247 | 1 | 1 | |
| | Merchant | 74 | 15 | 4.26 (2.39, 7.61) | 1.89 (0.78, 3.47) | 0.093 |
| | Others [a] | 9 | 22 | 0.35 (0.16, 0.78) | 1.15 (0.46, 2.68) | 0.089 |
| **Exposure to media** | Exposed | 297 | 136 | 4.49 (3.17, 6.35) | 3.69 (2.41, 5.65) | 0.001 |
| | Not exposed | 72 | 148 | 1 | 1 | |
| **Parity** | 1 | 37 | 37 | 1 | 1 | |
| | 2–4 | 220 | 216 | 1.02 (0.62, 1.67) | 1.12 (0.16, 7.74) | 0.072 |
| | >4 | 112 | 31 | 3.61 (1.97, 6.62) | 2.94 (0.26, 3.94) | |
| **Family size** | 2–3 | 41 | 43 | 1 | 1 | |
| | 4–6 | 219 | 214 | 1.07 (0.67, 1.71) | 1.3 (0.7, 2.43) | 0.078 |
| | $\geq 7$ | 109 | 27 | 4.23 (2.32. 7.72) | 0.69 (0.29, 1.63) | 0.117 |
| **History of abortion** | Yes | 70 | 9 | 7.54 (5.6, 9.6) | 11.6 (3.3, 14.6) | 0.047 |
| | No | 290 | 281 | 1 | 1 | |
| **Time to reach health facility** | <1 hour | 237 | 194 | 3.87 (2.88, 5.37) | 4.58 (3.05, 6.88) | 0.042 |
| | $\geq 1$ hour | 132 | 90 | 1 | 1 | |
| **Experienced danger signs** | Yes | 99 | 20 | 4.84 (2.9, 8.1) | 7.3 (3.9, 13.6) | 0.000 |
| | No | 270 | 264 | 1 | 1 | |
| **Decision-making power** | Higher | 314 | 181 | 3.25 (3.23, 4.73) | 1.17 (0.68, 1.98) | 0.237 |
| | Lower | 55 | 103 | 1 | 1 | |

Notes

[a] = student and daily laborer.

**Abbreviations:** AOR, adjusted odds ratio; COR, crude odds ratio; CI, confidence interval; 1, reference category.

**Table 6. Bi-variable and multivariable logistic regression analysis of factors associated with attitude towards antenatal care uptake among women delivered at home in rural Sehala Seyemit district, northern Ethiopia, 2020 (n = 653).**

| Variables | Category | Attitude of women | | COR (95% CI) | AOR (95%CI) | P-value |
|---|---|---|---|---|---|---|
| | | Favorable | Unfavorable | | | |
| **Age in year** | ≤ 20 | 62 | 8 | 1 | 1 | |
| | 21–30 | 309 | 131 | 0.3 (0.14, 0.65) | 0.95 (0.39, 2.3) | 0.066 |
| | >31 | 120 | 23 | 0.67 (0.28,1.59) | 0.56 (0.2, 1.54) | 0.063 |
| **Husband involvement** | Involved | 316 | 47 | 4.42 (3.0, 6.5) | 1.7 (0.77, 3.74) | 0.258 |
| | Not involved | 175 | 115 | 1 | 1 | |
| **Knowledge of ANC** | Adequate | 329 | 40 | 6.2 (4.14, 9.27) | 2.2 (1.26, 3.71) | 0.000 |
| | Inadequate | 162 | 122 | 1 | 1 | |
| **Decision-making power** | Higher | 436 | 59 | 13.84 (9.04,21.2) | 8.3 (4.8, 13.83) | 0.023 |
| | Lower | 55 | 103 | 1 | 1 | |
| **History of danger signs** | Yes | 108 | 11 | 3.87 (2.0, 7.4) | 2.36 (1.04, 5.37) | 0.109 |
| | No | 383 | 151 | 1 | 1 | |
| **Delivery assistant** | HEWs | 256 | 128 | 2.24 (1.14, 4.4) | 2.3 (1.1, 5.1) | 0.001 |
| | TBA | 225 | 29 | 0.21 (0.11, 0.41) | 0.64 (0.29, 1.41) | 0.079 |
| | Family | 10 | 5 | 1 | 1 | |
| **Women's education** | Unable to read and write | 256 | 128 | 1 | 1 | |
| | Able to read and write | 225 | 28 | 3.88 (2.49, 6.03) | 1.43 (0.73, 2.88) | 0.074 |
| | Primary education | 10 | 5 | 1.0 (0.33, 2.99) | 0.61 (0.15, 2.52) | 0.054 |
| **Media exposure** | Yes | 386 | 47 | 8.99 (6.0, 13.4) | 2.27 (1.30, 3.97) ** | 0.000 |
| | No | 105 | 115 | 1 | 1 | |

**Abbreviations:** AOR, adjusted odds ratio; COR, crude odds ratio; CI, confidence interval; HEWs, health extension workers; 1, reference category.

### Factors associated with women's attitude ANC uptake

On the multivariable logistic regression analysis, knowledge of ANC, household decision-making power, delivery assisted by health extension workers, and exposure to mass media were significantly associated with a favorable attitude of women towards ANC uptake.

The odds of having a favorable attitude towards ANC uptake among women who had adequate knowledge of ANC were two times higher (AOR = 2.2; 95% CI: 1.26, 3.71) as compared to women who had poor knowledge of antenatal care. The present study revealed that women who had higher household decision-making power were 8 times (AOR = 8.3; 95% CI: 4.8, 13.83) more likely to have had a favorable attitude towards ANC uptake as compared to women who had less decision-making power. Similarly, the likelihood of having a favorable attitude towards ANC uptake among women who were assisted by health extension workers for their recent delivery was two times higher (AOR = 2.3; 95% CI: 1.1, 5.1) compared with women who were assisted by family members. Lastly, those women who were exposed to mass media were 2.27 (AOR = 2.27; 95% CI: 1.30, 3.97) times more likely to have had a favorable attitude towards ANC uptake as compared to those women who didn't have media exposure (Table 6).

### Discussion

The present study assessed comprehensive knowledge of ANC, attitude towards its uptake, and associated factors among women who gave childbirth at home in rural Sehala Seyemit district, northern Ethiopia. More than half of women had adequate knowledge of ANC, and about three-fourths of them had a favorable attitude towards its uptake.

In this study, women's knowledge of ANC is lower than in previous studies conducted in Saudi Arabia [29], Libya [33], Ghana [5], and other studies conducted in Ethiopia, including Mizan [4] and Fiche towns [28]. This inconsistency might be because of differences in the study period, socio-demographic characteristics, study population, and study setting. This study includes the rural population, in which access to health-related information and the level of recognition and understanding is expected to be lower. As supported by another study, rural women were less likely to use maternal health services due to limited access to information [48]. In addition, 95.7% of the study participants in the Saudi Arabia and 64% in Libya studies have attained college and above education. However, almost all (97.7%) of the study participants in this study have no formal education. Empirical evidence showcases that higher education attainment has been associated with increased women's knowledge of ANC and maternal health service utilization [49,50]. Moreover, the disparity may be a result of the tool we used to measure the outcome variable; where most of the previously conducted research measured the outcome variable using a few yes/no questions.

Women's knowledge of ANC was, however, higher than findings from Malaysia [51] and Nigeria [42]. We might expect the disparity is differences in socio-demographic characteristics and the time gap. An alternative justification might be efforts have been made to enhance women's knowledge of any healthcare-related services in Ethiopia.

The current study has shown that the attitude of women towards ANC uptake was 75.2%, which was lower than findings in Libya [33] and Saudi Arabia [29]. This inconsistency might be due to socio-demographic differences like educational status and residence. Hence the studies from Libya and Saudi Arabia reported that over two-thirds and more than 90% of the participants were secondary and above by education, respectively. However, almost all the study participants had no formal education during the current study. Evidence showed that educated women were more likely to use maternal healthcare services [5,22]. On the other hand, in the present study, attitude towards antenatal care was higher than in previous studies conducted in Nigeria [42] and Mizan town, Ethiopia [4]. The variation might be for the time gap and increased awareness of women about maternal health services over time.

This study determined that the odds of adequate knowledge of ANC in the age groups of 20–30 and above 30 years old were seven and two times higher as compared to those women ≤ 20 years old, respectively. This finding contradicts a previously conducted study in Ghana, in which older women were less likely to have had adequate knowledge of antenatal care [5]. This discrepancy might be a difference in the study population, in which we gathered the data. Evidence has determined that older maternal age is associated with maternal healthcare service utilization [22]. Subsequently, if older maternal age enhances the practice of attending healthcare services, women may get knowledge concerning maternal and children's health.

Media exposure is an important predicting factor for women to have sufficient knowledge of ANC. Accordingly, women who had mass media exposure were nearly four times more likely to be knowledgeable as compared to those women who hadn't been exposed to mass media. The possible justification could be the health of women and children is a prime concern both at the national and global levels, thus maternal and children's health-related information may have been disseminated through mass media. Evidence also supports that exposure to mass media increases the likelihood of maternal health service utilization [52,53].

This study likewise found that those women who traveled less than an hour to reach health facilities were 4.58 times more likely to have had adequate knowledge of ANC compared with those women who traveled over an hour. The possible explanation could be that women far from health facilities are less likely to use healthcare services since they became tired because of the long travel. Besides, the far residence from health institutions may deter health-seeking

information, less the chance of getting health professional advice, and education. Studies have shown that longer distance impedes maternal and children's healthcare service utilization [54,55]. The current study indicated that women who experienced obstetric danger signs in their recent pregnancy were seven times more likely to be knowledgeable as compared to their counterparts. Finding supports that having ANC visits increase women's knowledge of obstetric danger signs [56]. Again, women with a history of obstetric danger signs may visit health facilities to receive treatment, and realizing the consequences can be dangerous, as a result, they can hear more about the danger signs of pregnancy and ANC.

Similarly, the odds of having adequate knowledge of ANC were 11.6 times higher among women who had a history of abortion as compared to those women who haven't had an abortion. If a woman has had an abortion before, she will likely have adequate antenatal care for the worry of recurrence and better outcomes of the current pregnancy. Our findings revealed that women who had adequate knowledge of ANC were 2.2 times more likely to have had a positive attitude towards antenatal care uptake. Knowledgeable women have a better understanding of maternal health services and the bad outcomes of not using the services. It is because knowledge is a roadmap for any process and act in healthcare practice.

Household decision-making power is another significant factor associated with a favorable attitude towards ANC uptake. Women who had higher decision-making power were 8 times more likely to have had a positive attitude towards antenatal care uptake as compared to women who had less decision-making power. If women are involved in any decision-making process, they are highly likely to use maternity and children's health services, so the knowledge and attitudes of women will be ideal. Research has shown that women's decision-making power strengthens the uptake of maternal health services [57] and women's good practices for neonatal illness [58].

The present study depicts that the likelihood of retaining a favorable attitude towards ANC uptake among women who were assisted by HEWs for their recent childbirth was 2.3 times higher compared with women who were assisted by family members. Even if they gave birth at home, women who are assisted by HEWs might get good advice about reproductive health, ANC, and general health services. Studies implicate that HEWs play an outstanding role in maternal and child healthcare service utilization. They refer women to the health facilities, assist delivery at home, and request ambulances during the need for referrals [59,60]. Lastly, this study has shown that a favorable attitude towards ANC is significantly associated with mass media exposure. Those women who had mass media exposure were two times more likely to have had a favorable attitude towards ANC uptake as compared to women who didn't have mass media exposure. Media exposed women can further inquire about information on maternal and children's health and gain sufficient data regarding antenatal care and any family health. Furthermore, they can weigh the pros and cons, use maternal health services, and have a positive perspective.

## Limitations and strengths of the study

This study could have certain limitations for the readers and policymakers need to consider. First, since the study was self-reported, social desirability bias may have been introduced. However, a better way of understanding was given for the study participants to give factual information. Second, since we have included all women who gave birth in the last 2 years, recall bias might have been encountered. Third, it may not be possible to infer cause and effect relationship between the outcome and the explanatory variables due to the cross-sectional nature of the study. Lastly, the quantitative nature of the data might fail to explore the socio-cultural and behavioral determinants of health. Despite these limitations, this study will have

paramount public health importance in the context of home delivery and rural settings. It will aspire to trail knowledge in this area and can be used as a benchmark for future studies as there was a dearth of similar studies reaching out to home-delivered women in Ethiopia. In addition, the use of a probability sampling technique, adequate sample size (i.e. representative), and community-based study could be the strength of this study.

## Conclusion

Despite having an optimum attitude towards ANC uptake, the majority of women had insufficient knowledge of antenatal care. Older age, media exposure, less travel time to reach the health facility, history of abortion, and women who experienced obstetric danger signs were predictors of knowledge of ANC. Likewise, adequate knowledge of ANC, higher household decision-making power, mass media exposure, and delivery assisted by HEWs were factors independently associated with a favorable attitude of women towards ANC uptake. The policymakers, program managers, and healthcare providers play a role in awareness creation in the communities for policies prioritizing those disadvantaged groups, have access to the health care which all mothers deserve. In addition, concerned bodies need to reach out to the rural community towards ensuring women's empowerment and engaging women in household decision-making matters, arranging easy transportation access, and strengthening the health extension program. This will further enhance women to use of maternal health services and keep them knowledgeable about ANC and other services related to women and child health. Further qualitative studies might be needed to explore the socio-cultural and behavioral determinants of health, which may significantly affect the knowledge and attitude of women towards ANC and institutional delivery utilization.

## Supporting information

**S1 File. Detail of knowledge measurement.**
(DOCX)

**S2 File. English version of the questionnaire.**
(DOCX)

## Acknowledgments

We would like to thank Debre Berhan University for providing the study ethical clearance. Our gratitude also goes to all data collectors and study participants. We are also glad to Sehala Seyemit Woreda Health Office for writing the permission letter.

## Author Contributions

**Conceptualization:** Azmeraw Ambachew Kebede.

**Data curation:** Azmeraw Ambachew Kebede, Birhan Tsegaw Taye.

**Formal analysis:** Azmeraw Ambachew Kebede, Birhan Tsegaw Taye, Kindu Yinges Wondie.

**Investigation:** Azmeraw Ambachew Kebede.

**Methodology:** Azmeraw Ambachew Kebede, Birhan Tsegaw Taye.

**Validation:** Azmeraw Ambachew Kebede, Birhan Tsegaw Taye.

**Visualization:** Azmeraw Ambachew Kebede, Birhan Tsegaw Taye, Kindu Yinges Wondie.

**Writing – original draft:** Azmeraw Ambachew Kebede, Birhan Tsegaw Taye, Kindu Yinges Wondie.

**Writing – review & editing:** Azmeraw Ambachew Kebede, Birhan Tsegaw Taye.

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
