## [Decision Letter · Decision Letter 0]

1 Dec 2021

PONE-D-21-01426Comprehensive Knowledge of Antenatal Care, Attitude towards Its Uptake and Associated Factors among Home-Delivered Women in Rural Sehala Seyemit District, Northern Ethiopia: A Community-Based Cross-Sectional StudyPLOS ONE

Dear Dr. Kebede,

Thank you for submitting your manuscript to PLOS ONE. After careful consideration, we feel that it has merit but does not fully meet PLOS ONE’s publication criteria as it currently stands. Therefore, we invite you to submit a revised version of the manuscript that addresses the points raised during the review process.

I would like to sincerely apologise for the delay you have incurred with your submission. It has been exceptionally difficult to secure reviewers to evaluate your study. We have now received three completed reviews; their comments are available below. The reviewers have raised significant scientific concerns about the study that need to be addressed in a revision.

Please revise the manuscript to address all the reviewer's comments in a point-by-point response in order to ensure it is meeting the journal's publication criteria. Please note that the revised manuscript will need to undergo further review, we thus cannot at this point anticipate the outcome of the evaluation process.

We look forward to receiving your revised manuscript.

Kind regards,

Miquel Vall-llosera Camps

Senior Editor

PLOS ONE

Journal Requirements:

Reviewers' comments:

Reviewer's Responses to Questions

**Comments to the Author**

1. Is the manuscript technically sound, and do the data support the conclusions?

Reviewer #1: Yes

Reviewer #2: Partly

Reviewer #3: Partly

2. Has the statistical analysis been performed appropriately and rigorously? 

Reviewer #1: Yes

Reviewer #2: Yes

Reviewer #3: Yes

3. Have the authors made all data underlying the findings in their manuscript fully available?

Reviewer #1: Yes

Reviewer #2: Yes

Reviewer #3: Yes

4. Is the manuscript presented in an intelligible fashion and written in standard English?

Reviewer #1: Yes

Reviewer #2: Yes

Reviewer #3: Yes

5. Review Comments to the Author

Reviewer #1: The Authors have done a good work on this important area of antenatal care. However, the are minor corrections to consider: 1. Table 1, Education status for the woman and husband should be in meaningful educational levels. Someone either know how to read and write or does not know. Primary education is a meaningless category there.

2. The variable, "type of transport at the time of emergency" typically apply when health facilities or locations are studied or when the study dwells on transport type available to women who experienced an emergency. As it applies here all participants from the same location should ideally give the same answer.

3. Study limitations should include an inherent bias in memorizing due to the retrospective nature of the study

4. Table 3 looks like containing raw data. It can be summarized with analytical data. This current table can be uploaded as additional data file.

Reviewer #2: Title

In line with the study objectives and the scope of the study, I would recommended the author revise the title to “Factors Associated with Comprehensive Knowledge of Antenatal Care and Attitude Towards ANC Uptake among Home-Delivered Women In Rural Sehala Seyemit District, Northern Ethiopia: A Community-Based Cross Sectional Study”.

Abstract

Introduction: you need to tighten your introduction. Transition from the first sentence to the second is weak. You talked about Ethiopia having the highest Maternal Mortality then straight to the gap of your study.

Methodology: Significant variables are not determined by adjusted odds ratio. Revise this sentence “Statistically significant association of variables was determined based on adjusted odds ratio with a 95% confidence interval and p-value of ≤0.05”. In addition, the word statistically significant is tautology. You can use “significantly associated”. Change bivariable to bivariate and multivariable to Multivariate. Do the same in the main work.

Results: Please take following out from the first sentence (95% CI; 52.6, 60.6), (95% CI; 71.8, 78.5). Is a univariate reporting and this does not need them. They are more relevant for multivariate reporting

Introduction

The introduction was well written, however, they need to revise the introduction to include the context of home delivery. Home delivery which is a major part of the work was mentioned once in the introduction (paragraph 7). The introduction should capture the main ideas of the work; ANC knowledge, attitude towards ANC uptake and home-based delivery.

I believe the study population (home based delivery respondent) has an impact on the findings of the study. Please justify why you choose home-based population. Why is it important to study knowledge on antenatal care and attitude towards ANC uptake among women who delivered at home?

Paragraph 3, first sentence, kindly get a more recent statistics. 2017 global maternal mortality rate of 295, 000 is a bot old considering we are in 2021. The figure is almost 4 years and things have changed. Using 2019 or 2020 maternal mortality figure would be appropriate.

Paragraph 7, first sentence, please delete “elsewhere” and be specific on the studies. Include the region or countries that the studies were conducted.

What gap is the study filling? The problem statement is not clear and strong. The author should indicate the gap that the paper intends to fill. The authors’ claim that “most studies failed to apply logistic regression analysis to establish factors correlated with women’s knowledge of ANC and attitude towards its uptake” cannot be accepted. Because there are some studies in Ethiopia that have applied logistic regression to examine the factors associated with ANC uptake. Read Berhan and Mohammed (2016), systematic review and meta-analysis by Tesfalidet et al., (2019).

In addition, less emphasized was put on home delivery in the problem statement. This is where the paper should elaborate on home delivery.

Please revise the last paragraph and highlight the new thing(s) that the study intends to highlight.

Methodology

Study designs, setting and period

References should be provided to support some of the claims. For instance, there should be reference for the “total population” and the sentence “accessing health services in the district is difficult because of the lack of transportation to each “kebeles”

Sample size determination and sampling procedure

Women’s attitude towards ANC of 70.6% is from where? Be specific if it is a district figure or a national figure.

Please explain the steps involved in applying multistage sampling very well. What you have written is not clear to me.

Measurement of variables

Please include questions on the comprehensive knowledge of ANC.

Please provide details on how the knowledge of antenatal care was categorized. Because some of the responses were binary (yes and no) whilst others were multi-response. How did you treat questions with three or four responses as well as the binary response to arrive at the adequate and inadequate responses? Explain the process of arriving at the categorization well.

Provide justification for using the mean as the cutoff point for favorable/unfavorable and adequate/inadequate respectively. Is it because the data were skewed or it was symmetric or something else.

Is the Likert scale questions on women’s attitude an existing scale? If so, Kindly indicate the source and studies that have used the likert scale. If not, provide the justification for using the likert scale.

Why is it that only two independent variables thus husband involvement and Household decision making power were explained. Kindly explain how the rest of the independent variables were measured.

Data collection and instruments

The first sentence should read “The data collection tool was developed by reviewing literature (3, 4, and 23). The data were collected using a semi-structured questionnaire through face-to-face interviews.

Please include a statement in the data collection and instruments on how participants consented to be part of the study.

Was participants confidentiality and anonymity assured? If so, kindly be specific

Data Processing and analysis

The first paragraph, line 5 reads “Binary logistic regression analysis was fitted to identify statistically significant independent variables, and variables having a p-value of less than 0.2 were included in the multivariable logistic regression for controlling confounders”. Please explain why you used confidence level of 80%, thus p value less than 0.2 at the bivariate level.

Binary logistic regression analysis should be changed to bivariate analysis.

The type of regression run at the multivariate level should be specified.

Kindly provide reasons why you used p value less than 0.2 at the bivariate level and p value less than 0.05 at the multivariate level

Ethical Approval and Consent to Participate

The last sentence “Both oral and written informed consent was collected from each of the study members was obtained after a clear explanation of the aim of the study” should be revised.

Add if all the respondent signed informed consent form or some thumbprint.

Results

This sentence should be revised “Most (97.2%) of the participants were in marital union and 384 (58.8%) of them were unable to read and write by education”. Replace marital union with currently married and take away education at the end of the sentence. The sentence should now read “most (97.2%) of the participants were currently married and more than half (58.8%) of the participants were unable to read and write.

Show the p value for all the bivariate and multivariate variables.

Please consider revising the tables by following this format. What you have presented is a bit confusing. The univariate and bivariate analysis should be together in one table and show the p value for each variable. This can be seen by having the variables, percentage (frequency), and then knowledge of women on ANC (adequate and inadequate). The multivariate table should be in one Table

Knowledge of women

Variables Percentage (Frequency) Adequate Inadequate P Value

Media exposure

Exposed

Not Exposed

From Table 1, on family size, I am wondering if all the women had at least 2 or more family size, Was there not any one with one family size?

Change women’s educational level to women’s literacy because read and write are literacy indicators not education. The same with husband educational status

On type of transport at the time of emergency, what is the difference between on foot, traditional ambulance and ambulance?

The authors should run multicollinearity analysis

Discussion

Second paragraph first sentence, please be clear on the specific findings that were lower than the previous studies

Edit second paragraph.

Conclusion of the study

From what I read in the manuscript, I think your conclusion is not sound.

Again, what is the policy direction of your work, kindly suggest at least two policies from the findings of the study

Recommend a future studies from your findings

Limitation and strength of the study

Aside social desirability, what are the other limitations of the study.

What is novel about your studies? What new thing are you adding to knowledge? Kindly make these clear

Reviewer #3: Review comments:

I think, in general, the study is very interesting, providing important issues about maternal attitudes towards ANC, their knowledge, partner involvement. The authors highlighted crucial factors about seeking ANC care for women who delivered at home, the results were nicely reported, and discussion was in accordance with the aims. I have some comments, which are presented below:

1. It would be helpful for reader to understand the study population, if the authors described them in terms of general Ethiopia population – how do they differ from general population, if there is any specific characteristics which might provide different results, any factors affecting specifically study population?

It was mentioned in the manuscript p. 6. ‘Before the actual data collection, we did a pretest on 34 women at Ziquala Woreda which has similar sociocultural and living standards as the study population.’ – that’s good. Just to add probably one paragraph about general vs study population.

2. As mentioned in the methods section, all home deliveries are included in the study, and if I understood correctly, you chose 670 women among those who delivered home (and did not have complications)? If so, what is this proportion? How many were women who gave birth in recent two years, how many among those delivered at home and how many did not have complications?

3. It would be also nice if authors can include some information about financing ANC and delivery-related costs, how it works in the country? Every pregnant woman can have free 4 visits (as it says they still have 4 ANC visits rather than 8), is it free of charge or do they need to pay some part of it? Also, is facility-based delivery paid out-of-pocket or by government?

4. The aim of the study was: ‘The present study assessed comprehensive knowledge of antenatal care, attitude towards its uptake, and associated factors among home-delivered women in rural Sehala Seyemit district, northern Ethiopia.’ I think, the first two aims are fine, I have concerns about the third aim – associated factors among home-delivered women – for me, its more relevant to say when comparing women who delivered at home to those who delivered at the institution, but in this study only women who delivered at home were included. I would suggest modifying the last aim slightly and address it in a way like you have presented in the results, probably including ‘barriers to use institutional deliveries’ or something similar.

5. In discussion part it is mentioned that: ‘This inconsistency might be because of differences in the study period, socio-demographic-characteristics, study population, and study setting’ – that’s exactly my concern – how does this study population differ from general population in terms of socio-demographic characteristics, if you could add the information about it, probably in the methods section when describing study population.

6. I suggest explaining strengths and limitations in more details.

6. PLOS authors have the option to publish the peer review history of their article (what does this mean?). If published, this will include your full peer review and any attached files.

Reviewer #1: **Yes: **Projestine Selestine Muganyizi

Reviewer #2: No

Reviewer #3: No

---

## [Decision Letter · Decision Letter 1]

27 Apr 2022

PONE-D-21-01426R1

Factors associated with comprehensive knowledge of antenatal care and attitude towards its uptake among home-delivered women in rural Sehala Seyemit district, northern Ethiopia: a community-based cross-sectional study.

PLOS ONE

Dear Dr. Kebede,

Thank you for submitting your manuscript to PLOS ONE. After careful consideration, we feel that it has merit but does not fully meet PLOS ONE’s publication criteria as it currently stands. Therefore, we invite you to submit a revised version of the manuscript that addresses the points raised during the review process.

The manuscript has been evaluated by one reviewer, and his comments are available below.

The reviewer has raised a number of concerns that need attention. He requests improvements to the reporting of methodological aspects of the study, for example, regarding the study population and the measurement variables, revisions to the statistical analyses and conclusion.

Could you please revise the manuscript to carefully address the concerns raised?

We look forward to receiving your revised manuscript.

Kind regards,

Lorena Verduci

Staff Editor

PLOS ONE

Journal Requirements:

Reviewers' comments:

Reviewer's Responses to Questions

**Comments to the Author**

1. If the authors have adequately addressed your comments raised in a previous round of review and you feel that this manuscript is now acceptable for publication, you may indicate that here to bypass the “Comments to the Author” section, enter your conflict of interest statement in the “Confidential to Editor” section, and submit your "Accept" recommendation.

Reviewer #2: (No Response)

2. Is the manuscript technically sound, and do the data support the conclusions?

Reviewer #2: Yes

3. Has the statistical analysis been performed appropriately and rigorously? 

Reviewer #2: Yes

4. Have the authors made all data underlying the findings in their manuscript fully available?

Reviewer #2: Yes

5. Is the manuscript presented in an intelligible fashion and written in standard English?

Reviewer #2: Yes

6. Review Comments to the Author

Reviewer #2: Abstract

Introduction:

Comment: You need to be specific on the Sustainable development goal.

Kindy revise the statement “However, evidence regarding the hypothesized topic is scarce, particularly which addresses women who gave birth at home”. I am confused about the “hypothesized topic: which is hypothesis topic are you talking about here. Kindly go straight to the point and write for easy understanding.

Kindly reduce the background. It is too long

Methodology:

Comment: Kindly take pretested questionnaire from the sentence. The sentence should read “The data were collected using a semi-structured interview guide”

Introduction

In paragraph 3, you argued that pregnant women should have at least eight ANC contacts or visit to antenatal clinics (In 2016, the World Health Organization (WHO) recommends that all mothers should have at least eight ANC contacts to ensure better maternal and neonatal health) but in paragraph 6 you argued about four visits (In connection to this, there is a need to assess the knowledge and attitude of the conservative rural community towards ANC utilization to be in line with the WHO target, which says 90% of pregnant women should have at least four and above ANC contacts and give birth at health facilities by 2025)…. Kindly be consistent which is which???

Last paragraph “you need to tighten your gap well. The paragraph does not read well and I do not really get the message, what is the contribution of the paper to literature and policy direction?

Kindly specify the goal number for all the sustainable goals in the manuscript

Methodology

Study population

Kindly rewrite the sentence well. The sentence should read “all women who gave birth in the last two years in the selected “Kebeles” during the data collection period were selected to be part of the study”

Measurement of variables

Please provide details on how the knowledge of antenatal care was categorized. Because some of the responses were binary (yes and no) whilst others were multi-response. How did you treat questions with three or four responses as well as the binary response to arrive at the adequate and inadequate responses? Explain the process of arriving at the categorization well.

Is the Likert scale questions on women’s attitude an existing scale? If so, Kindly indicate the source and studies that have used the likert scale. If not, provide the justification for using the likert scale.

Data Processing and analysis

Kindly provide reasons why you used p value less than 0.2 at the bivariate level and p value less than 0.05 at the multivariate level. It is still not clear

Results

Please avoid using women and respondents interchangeably. Kindly use one

Third sentence..change two-thirds of them to two-thirds of the respondents

Delete the full stop after “Most (97.2%) of the study participants were currently married”

Kindly delete “(95 % CI; 52.6, 60.6)” from the section women’s comprehensive knowledge of ANC. Do the same for attitude towards ANC uptake.

Conclusion of the study

I don’t understand this statement “The results of this study are looking for relevant stakeholders and health policymakers who pay particular attention to women in rural areas, including advocation regarding maternal health services through mass media” The results are looking for who?? Please delete or rewrite

Your conclusion is still not sound.

Kindly rewrite this “Moreover, researchers should address the rural women, particularly those who gave childbirth at home and the reason for not using maternal health services should better be addressed through a qualitative study” How can researchers address

There are grammatical errors in the work. Kindly edit the work.

7. PLOS authors have the option to publish the peer review history of their article (what does this mean?). If published, this will include your full peer review and any attached files.

Reviewer #2: No

---

## [Decision Letter · Decision Letter 2]

10 Jul 2022

PONE-D-21-01426R2

Factors associated with comprehensive knowledge of antenatal care and attitude towards its uptake among home-delivered women in rural Sehala Seyemit district, northern Ethiopia: a community-based cross-sectional study.

PLOS ONE

Dear Dr. Kebede,

Thank you for submitting your manuscript to PLOS ONE. After careful consideration, we feel that it has merit but does not fully meet PLOS ONE’s publication criteria as it currently stands. Therefore, we invite you to submit a revised version of the manuscript that addresses the points raised during the review process.

As you can see in the report attached below, reviewer 2 still raises some minor concerns that need your attention. They request that you carefully justify your statistical approach and thoroughly copyedit your manuscript.

We look forward to receiving your revised manuscript.

Kind regards,

Dario Ummarino, PhD

Senior Editor

PLOS ONE

Journal Requirements:

Additional Editor Comments (if provided):

If you do not know anyone who can help you with copyediting, you may wish to consider employing a professional scientific editing service.  

Whilst you may use any professional scientific editing service of your choice, PLOS has partnered with both

American Journal Experts (AJE) and Editage to provide discounted services to PLOS authors. Both

organizations have experience helping authors meet PLOS guidelines and can provide language editing,

translation, manuscript formatting, and figure formatting to ensure your manuscript meets our submission

guidelines. To take advantage of our partnership with AJE, visit the AJE website (http://learn.aje.com/plos/)

for a 15% discount off AJE services. To take advantage of our partnership with Editage, visit the Editage

website (www.editage.com) and enter referral code PLOSEDIT for a 15% discount off Editage services. If

the PLOS editorial team finds any language issues in text that either AJE or Editage has edited, the service

provider will re-edit the text for free.

Reviewers' comments:

Reviewer's Responses to Questions

**Comments to the Author**

1. If the authors have adequately addressed your comments raised in a previous round of review and you feel that this manuscript is now acceptable for publication, you may indicate that here to bypass the “Comments to the Author” section, enter your conflict of interest statement in the “Confidential to Editor” section, and submit your "Accept" recommendation.

Reviewer #2: (No Response)

2. Is the manuscript technically sound, and do the data support the conclusions?

Reviewer #2: Yes

3. Has the statistical analysis been performed appropriately and rigorously? 

Reviewer #2: Yes

4. Have the authors made all data underlying the findings in their manuscript fully available?

Reviewer #2: Yes

5. Is the manuscript presented in an intelligible fashion and written in standard English?

Reviewer #2: Yes

6. Review Comments to the Author

Reviewer #2: Data Processing and analysis

I am still not clear… why didn’t you stick to one (thus, 0.25 throughout). Kindly provide reasons why you used a p-value less than 0.25 at the bivariate level and a p-value of less than 0.05 at the multivariate level. It is still not clear. I am tempted to believe that you used 0.25 to get more values to be significant in the bivariate section for the multivariate results. This to me is a force attempt.

Conclusion

This sentence should read “Stakeholders and health policymakers need to pay particular attention to women in rural areas, including advocation regarding maternal health services through mass media”.

You need to edit the work very well. There are grammatical errors in the work. Kindly edit the work.

7. PLOS authors have the option to publish the peer review history of their article (what does this mean?). If published, this will include your full peer review and any attached files.

Reviewer #2: **Yes: **Dr Martin Wiredu Agyekum

---

## [Decision Letter · Decision Letter 3]

29 Sep 2022

Factors associated with comprehensive knowledge of antenatal care and attitude towards its uptake among women delivered at home in rural Sehala Seyemit district, northern Ethiopia: a community-based cross-sectional study

PONE-D-21-01426R3

Dear Dr. Kebede,

We’re pleased to inform you that your manuscript has been judged scientifically suitable for publication and will be formally accepted for publication once it meets all outstanding technical requirements.

Kind regards,

Michael Wells

Academic Editor

PLOS ONE

Additional Editor Comments (optional):

The authors have made important and substantial revisions to their manuscript. The editor, along with the reviewers, agree that this manuscript should be accepted in its current format for publication.

Reviewers' comments:

Reviewer's Responses to Questions

**Comments to the Author**

1. If the authors have adequately addressed your comments raised in a previous round of review and you feel that this manuscript is now acceptable for publication, you may indicate that here to bypass the “Comments to the Author” section, enter your conflict of interest statement in the “Confidential to Editor” section, and submit your "Accept" recommendation.

Reviewer #2: All comments have been addressed

2. Is the manuscript technically sound, and do the data support the conclusions?

Reviewer #2: Yes

3. Has the statistical analysis been performed appropriately and rigorously? 

Reviewer #2: Yes

4. Have the authors made all data underlying the findings in their manuscript fully available?

Reviewer #2: Yes

5. Is the manuscript presented in an intelligible fashion and written in standard English?

Reviewer #2: Yes

6. Review Comments to the Author

Reviewer #2: The authors have addressed all my comments. I, therefore, recommend that the editor accepts the paper for publication.

Thank you

7. PLOS authors have the option to publish the peer review history of their article (what does this mean?). If published, this will include your full peer review and any attached files.

Reviewer #2: **Yes: **Dr. Martin Wiredu Agyekum

---

## [Editor Report · Acceptance letter]

3 Oct 2022

PONE-D-21-01426R3 

Factors associated with comprehensive knowledge of antenatal care and attitude towards its uptake among women delivered at home in rural Sehala Seyemit district, northern Ethiopia: a community-based cross-sectional study 

Dear Dr. Kebede:

I'm pleased to inform you that your manuscript has been deemed suitable for publication in PLOS ONE. Congratulations! Your manuscript is now with our production department. 

Kind regards, 

on behalf of

Dr. Michael Wells 

Academic Editor

PLOS ONE